# Programming cell growth into different cluster shapes using diffusible signals

Yipei Guo[1,2¤a]*, Mor Nitzan[1¤b], Michael P. Brenner[1]

**1** John A. Paulson School of Engineering and Applied Sciences, Harvard University, Cambridge, Massachusetts, United States of America, **2** Program in Biophysics, Harvard University, Boston, Massachusetts, United States of America

¤a Current address: Janelia Research Campus, Ashburn, Virginia, United States of America
¤b Current address: School of Computer Science and Engineering, Racah Institute of Physics, Faculty of Medicine, The Hebrew University of Jerusalem, Jerusalem, Israel
* yipeiguo@g.harvard.edu

**Data Availability Statement:** All codes used for generating and analyzing data can be found in the GitHub repository: https://github.com/yipeiguo/ProgrammingGrowth.

## Abstract

Advances in genetic engineering technologies have allowed the construction of artificial genetic circuits, which have been used to generate spatial patterns of differential gene expression. However, the question of how cells can be programmed, and how complex the rules need to be, to achieve a desired tissue morphology has received less attention. Here, we address these questions by developing a mathematical model to study how cells can collectively grow into clusters with different structural morphologies by secreting diffusible signals that can influence cellular growth rates. We formulate how growth regulators can be used to control the formation of cellular protrusions and how the range of achievable structures scales with the number of distinct signals. We show that a single growth inhibitor is insufficient for the formation of multiple protrusions but may be achieved with multiple growth inhibitors, and that other types of signals can regulate the shape of protrusion tips. These examples illustrate how our approach could potentially be used to guide the design of regulatory circuits for achieving a desired target structure.

## Author summary

Multicellular tissues exhibit a variety of shapes and spatial cellular arrangements. How can cells grow into clusters with certain structural features, and how complex do the corresponding growth regulatory mechanisms need to be? Here, we use a model where cells can secrete diffusible signals, and both the secretion rates of these signals as well as the growth rate of cells can be regulated based on their local chemical environment. The spatial profile of growth rate then drives any changes in the shape of the cluster as it expands. With this framework, we explore questions such as how easily (e.g., how many chemicals cells need to secrete) can we program the growth of a single protrusion, and when is it possible (or not possible) to grow multiple protrusions or a protrusion with a sharp tip. We find that the maximum number of protrusions, a measure of developmental complexity, can increase exponentially with the number of available signals. The mathematical

**Funding:** This research was funded by the National Science Foundation through DMS-1715477, MRSEC DMR-1420570, and ONR N00014-17-1-3029. MPB is an investigator of the Simons Foundation. M.N. acknowledges support from James S. McDonnell Foundation, Schmidt Futures, Israel Council for Higher Education, and the John Harvard Distinguished Science Fellows Program within the FAS Division of Science of Harvard University. Y.G. acknowledges support from the QBio fellowship awarded by the NSF-Simons Center at Harvard. The funders had no role in study design, data collection and analysis, decision to publish, or preparation of the manuscript.

**Competing interests:** The authors have declared that no competing interests exist.

framework we offer and the concrete predictions that follow could serve as useful guidelines for synthetic developmental biology experiments.

## Introduction

A fundamental goal of synthetic developmental biology is to understand how cells can be programmed to generate a desired spatial configuration. In order to achieve such a collective goal, individual cells must make appropriate local decisions depending on where they are in the cluster and the current global state of the system. However, cells do not have direct access to these quantities and can only sense their immediate surroundings. Global information that needs to be accessible for cellular decision making must therefore be encoded in their local environment.

Recent advances in genetic engineering technologies have made it possible to encode desired sets of rules within the genetic programs of cells, and these have been used to create distinct spatial patterns [1–5]. For example, the use of a synthetic notch receptor system to encode changes in the expression levels of cadherin molecules (in 'receiver cells' engineered with receptors that trigger downstream cellular responses when activated) upon contact with another cell type ('sender cells' engineered with ligands on cell surface) led to self-organization of clusters with distinct spatial arrangements of different cell types [3, 6]. In addition to juxtacrine signaling, i.e. signaling through direct cell-cell contact (ligand/receptor systems), cells can also be engineered to communicate via diffusible signals [4, 5, 7]. In particular, a graded pattern of signaling activity was obtained by culturing engineered Hedgehog-responding cells next to engineered Hedgehog-secreting cells [4], and Turing-like patterns were generated by reconstituting an activator-inhibitor circuit of two diffusible molecules [5].

However, besides spatial patterns within a cell cluster, another important aspect of spatial structure lies in the form of the overall tissue shape. While spatial patterning only involves the spatial arrangement of cells with different gene expression profiles, achieving a desired tissue morphology requires that differences in gene expression lead to downstream physiological effects such as changes in the growth/proliferation or death rates of cells. Synthetic circuits can potentially be engineered to produce these types of physiological outputs [1, 8]. Nevertheless, before any experimental attempt, it is useful to first ask what are the rules that would enable cells to grow into a desired shape.

The question of how tissue morphology emerges has been widely studied in many biological systems. Desired tissue shapes can arise from cell movement [9] or rearrangement [10–13], oriented cell divisions [14, 15], mechanical constraints [16, 17], differential growth rates [15–18], amongst others. These require cells to sense and respond to their local environment. For example, cells are known to sense and respond to mechanical cues [19, 20], and to interact with nearest neighbors through adhesion molecules on their surfaces [21, 22]. The ability to respond to local external chemical environments is also particularly important in many developmental processes, where some cells can secrete morphogens which diffuse in extracellular space to induce concentration-dependent responses in other cells [23, 24]. For example, in growing bird beaks, there is a region enriched with BMP growth factors near the tip of the beak known as the growth zone in which cells are actively proliferating [25]. In plant leaves, the chemical auxin activates growth while the chemical CUC2 is a growth repressor [26]. The concentration profiles of these chemicals therefore determine the spatial profile of growth rate across the leaf, and hence the resulting leaf shape [26]. The precise regulation of any specific biological phenomenon is typically very complex, involving a combination of different

mechanisms [14–17] or many types of signals or even multiple cell types, each following a different set of rules [11, 17]. One would also typically expect that the more classes of rules and regulation mechanisms cells have at their disposal, the larger the range of structures cells can grow into. However, when trying to grow a certain target structure in the lab, it is desirable to achieve a minimal working model, or to work with the minimal number of components. This requires an understanding of which elements are sufficient or insufficient for achieving a desired structural phenotype. Even within the context of a single cell type and a single type of regulatory mechanism, it is useful to understand how developmental complexity scales with the complexity of the cell-to-cell interaction rules [27].

In this paper, we attempt to address such questions by developing a framework for studying how different cluster shapes can emerge from cells regulating their growth rates based on their local chemical environment. Within our model, cells are allowed to secrete diffusible chemicals which can either directly regulate the growth rate of cells ('growth regulator'), or indirectly affect growth by changing the secretion rate or the effect of other growth regulators. We obtain the growth of the overall cell cluster by treating it as an incompressible fluid, an approach that has been used to model a wide variety of systems [28–37] including the development of multicellular tissues [28–34]. Within this simple class of models, we explore what is possible and how complex the regulatory schemes need to be when there is just a single cell type whose growth rate is regulated by diffusible morphogens alone. We find that (1) a single growth inhibitor can be used to grow a rod-like structure, (2) multiple growth regulators are required to grow multiple protrusions, and (3) the length and shape of each protrusion can be controlled using growth-threshold regulators. With these regulatory schemes, we also illustrate how our approach can be used to infer how developmental complexity (i.e. the range of achievable structures) scales with model complexity (i.e. the number of signals) for any given initial cluster. In particular, we find that the maximum possible number of protrusions increases exponentially with the number of growth inhibitors involved, and to control the growth of each set of protrusions, it is necessary to have an independent threshold regulator.

## Results

### Modeling diffusion-based morphogenesis

Suppose every cell has the potential to secrete $q$ different chemicals, with the secretion rate $\mu_i(\vec{c})$ of chemical $i$ potentially regulated by the local chemical environment $\vec{c}$ of the cell (Fig 1A):

$$\mu_i(\vec{c}) = \mu_{i,max}(\vec{c}) \prod_{j=1}^{q} H_{ij}(c_j | K_{ij}(\vec{c}), n_{ij}),\tag{1}$$

where $\mu_{i,max}$ is the maximum secretion rate of $i$, and $H_{ij}(c_j | K_{ij}, n_{ij}) = c_j^{n_{ij}}/(K_{ij}^{n_{ij}} + c_j^{n_{ij}})$ is the Hill function representing how chemical $j$ affects the secretion rate of $i$.

We consider a 2D cluster of cells in a liquid medium and assume that the growth rate is much slower than the secretion, degradation and diffusion rates, such that the spatial profiles of chemical concentrations satisfy the set of steady-state reaction-diffusion equations:

$$D_i \nabla^2 c_i + \mu_i(\vec{c}) - \gamma_i c_i = 0,\tag{2}$$

for $i = 1, \ldots, q$, where $D_i$ is the diffusion coefficient, $\gamma_i$ is the degradation rate of $i$, and we assume the concentrations vanish far from the cell cluster.

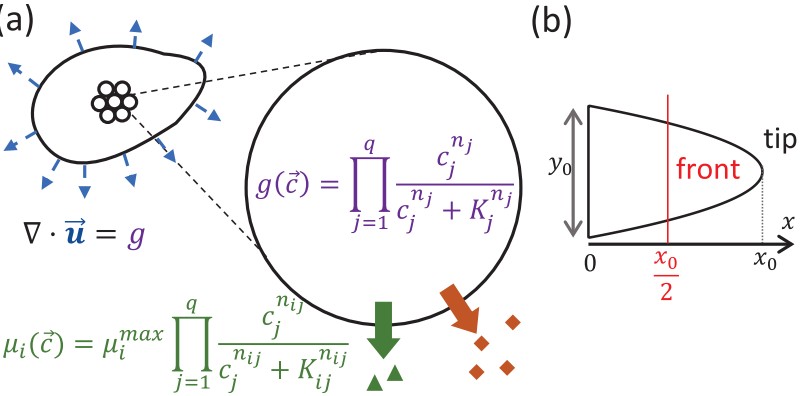

**Fig 1. Schematic of the model.** (a) We consider a cluster of cells, each having the potential to secrete $q$ different diffusible chemicals. The secretion rate $\mu_i(\vec{c})$ of each of these chemicals $i = 1, \ldots, q$, and the growth rate $g(\vec{c})$ of each cell depend on the local external chemical environment $\vec{c}$ of the cell. Given the spatial profile of growth rate inside the cluster, we can then solve for the velocity of cell flow $\vec{u}$, which specifies how the shape of the tissue changes over time (blue arrows). (b) We initialize the system with a 2-dimensional parabolic cluster with initial length $x_0$ and initial width $y_0$.

We also allow the chemicals to be *growth regulators* that regulate the growth and division rate $g$ of each cell (Fig 1A):

$$g(\vec{c}) = \prod_{j=1}^{q} H_j(c_j | K_j(\vec{c}), n_j), \tag{3}$$

where $H_j(c_j | K_j, n_j) = c_j^{n_j}/(K_j^{n_j} + c_j^{n_j})$ is the Hill function representing how chemical $j$ affects the growth rate of cells, and we have taken the maximum growth rate to be 1. A summary of the list of model parameters can be found in S1 Table.

In both the secretion and growth regulatory functions (Eqs 1 and 3), we take the Hill coefficients $n_{ij}$ and $n_j$ to be either 0 (when $j$ does not participate in the regulation), or very large in magnitude (negative when $j$ is an inhibitor, positive when $j$ is an activator) such that the Hill functions can be thought of as threshold functions.

Within a tissue, or a cellular cluster, the growth zone, which marks the region where cells can grow and divide, is determined by Eqs 1–3. To simulate the dynamics given the growth zone, we model the cell cluster as an incompressible cellular 'fluid' of constant density, such that any non-zero divergence of the velocity **u** of cell flow arises only because of the division of cells (Fig 1A):

$$\nabla \cdot \mathbf{u} = g, \tag{4}$$

with $\mathbf{u} = -\nabla P$ and we impose the condition that the pressure is $P = 0$ at the cluster boundary. For any $g(\mathbf{x})$, the velocity of the tissue along its boundary can then be obtained using boundary integral methods (see S1 Text) [38].

The equations above specify the dynamics obtained for any regulatory mechanism, including the number of chemicals involved and the role of each of these chemicals, i.e., how they affect the growth and secretion of other chemicals.

Given a fixed set of growth rules, the resulting cell cluster will also depend on the initial configuration. In particular, with an initial circular cell cluster, if there are no spontaneous instabilities, the chemical concentration profiles and hence the cluster will remain circularly symmetric as it grows. For more complex shapes to emerge, there must be some symmetry

breaking mechanism. In biological systems, the initial cluster shape is often determined by external conditions such as an external concentration gradient. Furthermore, during the early stages of development, the timing, the order and plane of cell divisions are typically highly regulated and possibly governed by a separate set of rules encoded within the cell. When trying to grow these tissues in the lab, one can imagine creating molds or patterned environments for initializing the arrangement of cells. However, our goal here is not to explore what can be achieved throughout the whole space of initial conditions. Instead, we will illustrate how our framework can be used to inform how regulatory circuits should be designed in order to grow desired structural features from a given initial cluster shape.

In the spirit of developing bird beaks, which have been shown to be shaped as conic sections with a localized growth zone near the tip [25], we choose to initialize the system with a 2-dimensional parabolic cell cluster of initial length $x_0$ and initial width $y_0$ (Fig 1B), and explore how protrusions can grow from the tip of this cluster. An initial elliptical cell cluster can also give rise to qualitatively similar structural features (see S1 Fig).

## Cluster becomes increasingly circular in the absence of growth regulation

We first consider a case with no growth regulation. Within our model, if all cells grow at the same rate, the cluster would expand in all directions and become more circular as it grows (Fig 2A). This is true even with a more elongated initial state (Fig 2B). Therefore, preferential elongation of the tissue in one direction can arise only when there is growth rate heterogeneity across the cluster, and this requires the use of a growth regulator.

In the next few sections, we ask how one can program the growth of protrusions using growth regulators, and how can the number and shape of protrusions be controlled with increasingly complex regulatory circuits. In particular, we will use concrete examples of growth regulation to explore how the range of achievable structures (namely, the number and type of protrusions) varies with the type of regulatory mechanism and the number of signals involved in the growth regulation. The regulatory schemes we explore in this paper do not exhibit spontaneous instability, and we return to this point in the Discussion section. Since we are interested in the growth of protrusions from the cluster tip, we will only focus on the growth of cells in the front half of the tissue (Fig 1B) and neglect any growth of cells on the opposite (vertical) end.

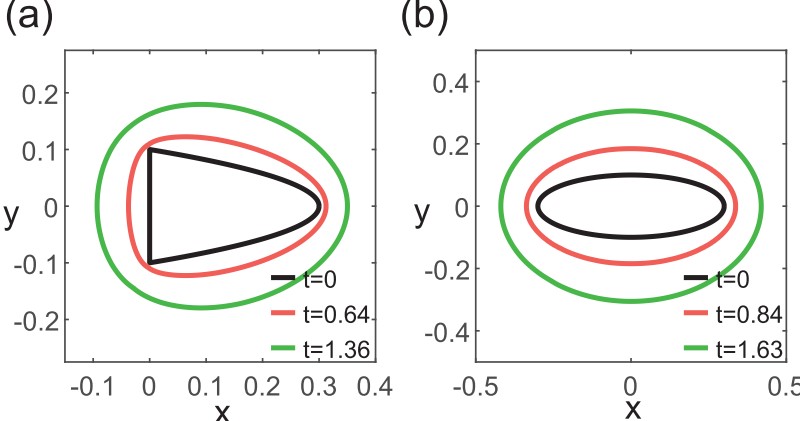

**Fig 2. Dynamics of cluster growth without growth regulation.** Cluster is initialized as (a) a parabolic cluster and (b) an elliptical cluster. As all cells grow at the same rate, an initial elongated cluster (black lines) becomes increasingly circular over time (black to pink to green).

There are two types of growth regulators: growth activators and growth inhibitors. In the simplest scenario where all cells secrete only a single growth regulator $X$ (and no other signals) at a constant rate, if $X$ is a growth activator, cells only grow when $X$ is above a threshold concentration. As the number of cells in the cluster increases, the regulator concentration $c_X$ increases and the number of growing cells increases. This constitutes a positive feedback loop, and the cluster becomes increasingly circular as it grows, analogous to the case where there is no growth regulation. In the remainder of the paper, we will therefore focus on regulatory mechanisms which involve growth inhibition rather than growth activation. Nevertheless, we show in the Supporting information an example of how a regulatory mechanism involving a growth activator can be used to grow a protrusion if cells secrete another chemical that inhibits the secretion of the growth activator (see S2 Text and S2 Fig).

## Single growth inhibitor can give rise to an elongating rod-like structure

We first consider the scenario where all cells secrete only a single growth inhibitor $X$ at a constant rate $\mu_X$. The inhibitor concentration $c_X$ satisfies the following non-dimensional equation:

$$\tilde{\nabla}^2 \tilde{c}_X + \tilde{\mu}_X - \tilde{c}_X = 0, \tag{5}$$

where the rescaled length scale $\tilde{x} = \sqrt{\frac{\gamma_X}{D_X}}x$, the rescaled inhibitor concentration $\tilde{c}_X = \frac{c_X}{K_{gX}}$ with $K_{gX}$ being the threshold concentration above which cells stop growing (i.e. $g(\tilde{c}_X) = H_X(\tilde{c}_X|1, -100)$ in Eq 3), and the effective secretion rate $\tilde{\mu}_X = \frac{\mu_X}{\gamma_X K_{gX}}$.

Given any initial cluster size, the subsequent dynamics are therefore determined only by a single parameter $\tilde{\mu}_X$, with the growth zone being the region where $\tilde{c}_X \leq 1$ (Fig 3A). For sufficiently high values of $\tilde{\mu}_X$ (such that $\tilde{c}_X > 1$ everywhere within the cluster), no cells can grow. For intermediate values of $\tilde{\mu}_X$, we find that the growth zone is localized near the tip of the parabolic cluster, and the size of the initial growth zone increases with decreasing $\tilde{\mu}_X$ (Fig 3B). With the growth zone at the tip, the cluster grows a rod-like structure regardless of the size of the growth zone (Fig 3B). In the long time limit, the chemical environment at the tip and hence the size of the growth zone stays approximately constant during growth (Fig 3C and 3D), with this steady-state size also decreasing with $\tilde{\mu}_X$ (Fig 3D). For low values of $\tilde{\mu}_X$ (such that $\tilde{c}_X < 1$ everywhere within the cluster), no cells will be inhibited. As the cluster expands in all directions, $c_X$ increases, and some of the cells will eventually become inhibited. If this occurs when the tissue is sufficiently elongated, it is possible for the growth zone to be localized at the tip of the cluster (like in the case of intermediate values of $\tilde{\mu}_X$), at which point a protrusion can emerge (S3 Fig).

Such a mechanism can therefore potentially be used for growing rod-like extensions. However, such an extension can grow indefinitely, and other mechanisms or external conditions are required for the cluster to stop growing. The termination of protrusion growth can occur, for example, when cells encounter depletion in nutrients, oxygen or other signals necessary for growth.

This result also implies that within our model and for an initial parabolic cell cluster, a single growth inhibitor alone cannot give rise to anything other than a single rod-like protrusion. We therefore ask how more complex shapes can develop with the use of more chemical signals. In the remainder of the paper, we will focus on two specific features: (a) the growth of multiple protrusions and (b) the growth of a protrusion whose width decreases over time (resembling a bird beak). The latter would also provide a mechanism for the autonomous termination of a protrusion (without the need for other external factors).

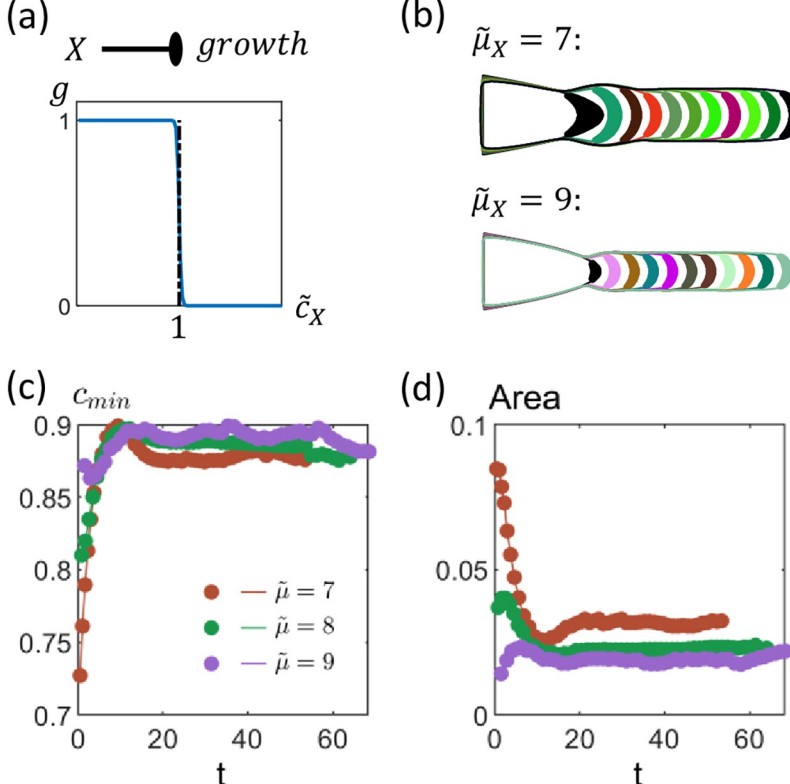

**Fig 3. Dynamics of a parabolic 2-D cluster with cells secreting a single growth inhibitor.** (a) We treat $X$ as a growth inhibitor (as indicated by the elliptical arrowhead), such that cells can only grow if the concentration of $X$ is below a threshold i.e. $\tilde{c}_X < 1$. (b) Only cells at the tip of the tissue can grow, with the initial size of the growth zone (black region) decreasing with increasing effective secretion rate of inhibitor $\tilde{\mu}_X$. As cluster grows, a rod-like extension emerges. The different colored regions represent the growth zones at different times. (c) Rescaled inhibitor concentration $\tilde{c}_{X,min}$ at the tip initially increases but reaches a steady-state level where it stays approximately constant. (d) Area of growth zone eventually stays approximately constant as the rod-like extension grows. The steady-state area decreases with $\tilde{\mu}_X$. [Other parameters: initial tissue length $\tilde{x}_0 = 1.5$, initial tissue width $\tilde{y}_0 = 1$].

## Growing multiple protrusions with multiple growth inhibitors

In order for multiple protrusions to develop, there has to be multiple peaks in the normal velocities along the cluster boundary. One way for this to occur is for the tissue to have multiple growth zones—these do not necessarily need to be present at the initial state as long as there is potential for these distinct growth zones to emerge during growth.

Within the framework of our model, a chemical can either influence the secretion rate of other chemicals (Eq 1), affect growth rate regulation by other chemicals through the growth threshold or is itself a growth regulator (Eq 3). We find that since the concentration of growth regulators most directly determines whether a cell can grow, including additional growth regulators can change growth zone shapes most drastically. Nevertheless, for each additional growth regulator to provide spatial information that the other growth regulators could not have provided, the secretion of these additional growth regulators must be regulated such that not all cells are producing the same set of growth regulators.

A simple mechanism for increasing the number of possible protrusions (through introducing the potential for growth zones to split) is to allow cells within a growth zone to produce an

additional growth inhibitor. We will first illustrate how this works with two growth inhibitors, before generalizing to the scenario of having multiple growth inhibitors.

**It is possible to grow up to three protrusions with just two growth inhibitors.** We consider here the case where $X$, in addition to being a growth inhibitor, also inhibits the secretion of a second growth inhibitor $Y$ (Fig 4A). This implies that $Y$ is only produced when $X$ is below some threshold level $c_X < K_s$ (Fig 4A), i.e.

$$\mu_Y(c_X) = \mu_{Y0}H_{YX}(c_X|K_s, -100).$$ (6)

The steady-state concentrations therefore satisfy the following set of non-dimensional equations:

$$\tilde{\nabla}^2\tilde{c}_X + \tilde{\mu}_X - \tilde{c}_X = 0$$
$$\tilde{\nabla}^2\tilde{c}_Y + \tilde{\mu}_Y(\tilde{c}_X) - \gamma_r\tilde{c}_Y = 0$$ (7)

where the rescaled length scale $\tilde{x} = \sqrt{\frac{\gamma_X}{D_X}}x$, rescaled concentrations $\tilde{c}_X = \frac{c_X}{K_{gX}}$, $\tilde{c}_Y = \frac{c_Y}{K_{gY}}$ and $\tilde{K}_s = \frac{K_s}{K_{gX}}$, effective secretion rates $\tilde{\mu}_X = \frac{\mu_X}{\gamma_X K_{gX}}$ and $\tilde{\mu}_Y = \tilde{\mu}_{Y0}H_{YX}(\tilde{c}_X|\tilde{K}_s, -100)$ with $\tilde{\mu}_{Y0} = \frac{\mu_{Y0}}{\gamma_X K_{gY}}\frac{D_X}{D_Y}$, and the rescaled $Y$ degradation rate $\gamma_r = \frac{\gamma_Y}{\gamma_X}\frac{D_X}{D_Y}$.

The corresponding growth condition is given by:

$$g(\tilde{c}_X, \tilde{c}_Y) = H_X(\tilde{c}_X|1, -100)H_Y(\tilde{c}_Y|1, -100).$$ (8)

Fixing the secretion rate of $X$ to be $\tilde{\mu}_X = 8$, we find that the inclusion of $Y$ allows the growth zone to take on different shapes (Fig 4B, 4C and 4D), and this influences the resulting tissue morphology (Fig 4E, 4F and 4G). The shape of the growth zone depends on the values of $\tilde{\mu}_{Y0}$ and $\tilde{K}_s$ (Fig 4B, 4C and 4D). In the low $\tilde{\mu}_{Y0}$ limit, the growth zone is determined solely by $c_X$. As $\tilde{\mu}_{Y0}$ increases, some of the cells that were not inhibited by $X$ may now be inhibited by $Y$. This reduces the size of the growth zone. The way in which the shape of the growth zone changes with increasing $\tilde{\mu}_{Y0}$ depends on the secretion threshold $\tilde{K}_s$ (Fig 4B, 4C and 4D). This is because the growth zone starts being depleted at the point where $c_Y$ is the highest (S4 Fig).

In particular, if cells secrete $Y$ only when their growth is not inhibited by $X$ ($\tilde{K}_s = 1$), the growth zone is depleted from the center of the original growth zone (Fig 4C). The presence of this 'hole' changes the velocity profiles along the boundary, leading to multiple peaks in velocity along the boundary. This in turn gives rise to a pair of protrusions extending from opposite sides of the tissue, in addition to the middle tip protrusion (Fig 4F). The tip protrusion eventually stops growing as $c_X$ (which includes contributions from cells in both side protrusions) increases (S5 Fig). Depending on the value of $\tilde{\mu}_Y$, which determines the initial fraction of growing cells and hence how fast $c_X$ increases, the tip protrusion can reach different lengths before it stops growing (Fig 4F). However, the side protrusions continue to grow because the increase in number of cells (and hence total production of $X$) is offset by these protrusions getting further away from the main bulk of the cluster.

This mechanism shows that having two growth inhibitors can give rise to a maximum of 3 protrusions. This is nevertheless only an upper bound. If $\tilde{K}_s$ is sufficiently large such that the growth zone depletes from the boundary closer to the bulk of the tissue (Fig 4D), there will just be a single growth zone and hence a single protrusion (Fig 4G). Furthermore, even with multiple distinct growth zones in the initial state, it is possible for the cluster to grow a single rod-like structure similar to when cells only produce $X$ (Fig 4E).

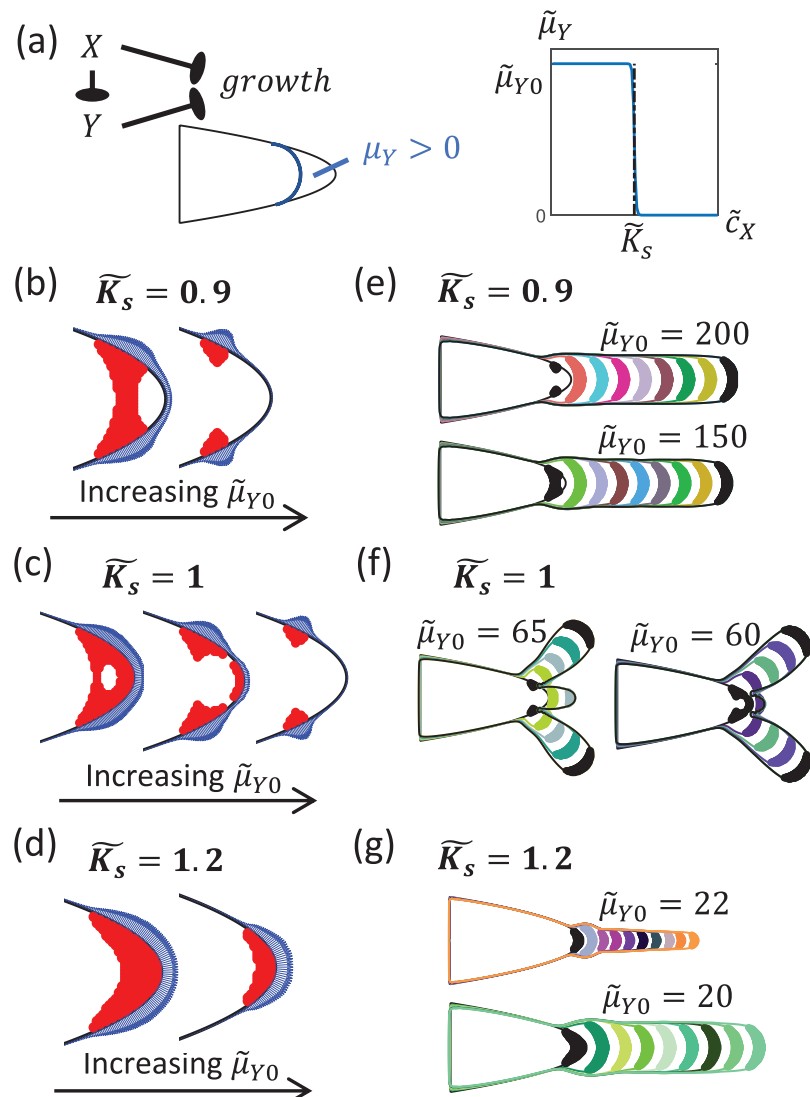

**Fig 4. Dynamics of cluster when cells secrete two growth inhibitors.** (a) We consider here the case where both $X$ and $Y$ inhibits growth. In addition, $X$ also inhibits the production of $Y$, such that its secretion rate $\tilde{\mu}_Y(\tilde{c}_X)$ is a threshold function with $Y$ only produced near the tip of the cluster where $\tilde{c}_X < \tilde{K}_s$. (b-d) The initial shape of the growth zone (in red) depends on the values of secretion threshold $\tilde{K}_s$ and $\tilde{\mu}_Y$ through concentration profile of $Y$ (S4 Fig). The blue regions consist of arrows perpendicular to the tissue boundary, with the length of the arrows proportional to the boundary velocity at that point. (b) When $\tilde{K}_s = 0.9$, the region of the growth zone closer to the tissue tip stops growing first as $\tilde{\mu}_{Y0}$ is increased (left: $\tilde{\mu}_{Y0} = 150$, right: $\tilde{\mu}_{Y0} = 200$). (c) When $\tilde{K}_s = 1$, growth zone depletes from its center as $\tilde{\mu}_{Y0}$ increases (left: $\tilde{\mu}_{Y0} = 56$, middle: $\tilde{\mu}_{Y0} = 60$, right: $\tilde{\mu}_{Y0} = 65$). (d) When the ratio of secretion threshold to growth threshold $\tilde{K}_s = 1.2$, the growth zone shrinks from the left boundary as $\tilde{\mu}_{Y0}$ increases (left: $\tilde{\mu}_{Y0} = 20$, right: $\tilde{\mu}_{Y0} = 22$). The growth zone therefore remains attached to the tip of the tissue. (e-g) Cluster dynamics for the different values of $\tilde{K}_s$, with the different colored regions representing the growth zones at different time points (colors randomly generated). (e) When $\tilde{K}_s = 0.9$, the cluster grows a single rod-like protrusion even though there are initially multiple growth regions. (f) When $\tilde{K}_s = 1$, cluster can grow multiple protrusions (S5 Fig). (g) When $\tilde{K}_s = 1.2$, there is only a single growth zone and hence a single protrusion. [Other dimensionless parameters: $\tilde{\mu}_X = 8$, $\gamma_r = 1$.]

**The potential number of protrusions scales exponentially with the number of growth regulators.**   This same strategy can be repeated with more than 2 growth inhibitors. For example, if a third growth inhibitor $Z$ is produced within the existing growth zone (i.e. when both $X$ and $Y$ levels are low), this can again split the existing growth zones, doubling the

number of potential protrusions. This argument therefore implies that with $q \geq 2$ growth regulators, it is possible to get a maximum of $z_{max} = 3 \times 2^{q-2}$ protrusions. Equivalently, $q_{min} = \log_2(z/3) + 2$ is the minimum number of growth regulators we would need to get $z$ protrusions. Together, the potential number of protrusions scales exponentially with the number of growth regulators.

## Regulating the growth of individual protrusions using threshold-regulators

In addition to growing multiple protrusions, one may also wish to regulate the growth of each of them i.e. control how the width changes over time and for the cluster to stop growing by itself. We first ask how this can be achieved for a single protrusion, before discussing the general case of controlling multiple protrusions.

**A growth threshold regulator, together with a growth inhibitor, can give rise to a cone-like protrusion.** We saw previously that with a single growth inhibitor, the protrusion will grow with constant width. This implies that for its width to change over time, additional chemicals are required. A mechanism for the protrusion to grow a sharp tip is for the growth zone to shrink and the center of the growth zone to shift closer to the tip as the protrusion grows. The protrusion terminates once the growth zone vanishes. This is in fact what happens in the development of bird beaks [25].

Inspired by this, we ask here how such a phenomenon can arise. One way this could happen is if the growth threshold $K_{gX}$ of the growth inhibitor $X$ decreases over time. However, since any growth rule must be local, the time dependence of $K_{gX}$ must occur through the dependence on the chemical environment i.e. $K_{gX}(\vec{c})$. We therefore considered the possibility of having a second chemical $Y$ that reduces $K_{gX}$. For simplicity, we chose a linear function for this threshold regulation (Fig 5A):

$$K_{gX}(c_Y) = K_{gX0}(1 - ac_Y), \tag{9}$$

where $K_{gX0}$ is the baseline growth threshold when $Y$ is absent, and $a$ controls how strongly $Y$ regulates $K_{gX}$.

Nevertheless, we find that the secretion of this threshold regulator is not a sufficient condition for the growth zone to decrease in size—it is necessary for the secretion of $Y$ to be regulated such that only certain regions of the tissue are secreting $Y$. This is because if $Y$ is secreted at a constant rate by all cells, $c_Y$ at the tip will not increase as the tissue grows (just like for the growth inhibitor $X$). In this case, cells at the growing tip do not have any information of how far they are from the bulk of the tissue, and the cluster again grows a rod-like structure (Fig 5B).

In order for only certain regions of tissue to secrete $Y$, we allow the secretion rate of $Y$ to depend on the concentration of $X$ as in the case of having 2 growth inhibitors (Eq 6). The steady-state diffusion-reaction equations for this context are the same as if there were 2 growth inhibitors (Eq 7), except that the rescaled concentrations are now $\tilde{c}_X = \frac{c_X}{K_{gX0}}$, $\tilde{c}_Y = ac_Y$ and $\tilde{K}_s = \frac{K_s}{K_{gX0}}$, the effective secretion rates $\tilde{\mu}_X = \frac{\mu_X}{\gamma_X K_{gX0}}$ and $\tilde{\mu}_Y = \tilde{\mu}_{Y0} H(\tilde{c}_X | \tilde{K}_s, -100)$ with $\tilde{\mu}_{Y0} = \frac{a\mu_{Y0}}{\gamma_X} \frac{D_X}{D_Y}$.

For the growth zone to decrease in size, $c_Y$ at the tip needs to increase as the tissue grows. We find that an effective way for this to occur is for cells to secrete $Y$ only when $\tilde{c}_X < \tilde{K}_s$ (Fig 5C), and for $\tilde{K}_s < 1$ such that only a subset of growing cells (close to the tip of the tissue) are producing $Y$ (Fig 5D). As the tissue grows, a reduction in $c_X$ at the tissue tip causes the secretion region of $Y$ to increase in size, increasing $c_Y$ at the tip (S6 Fig). This increase in $c_Y$ reduces the size of the growth zone over time, giving rise to a narrowing of the protrusion (Fig 5D and

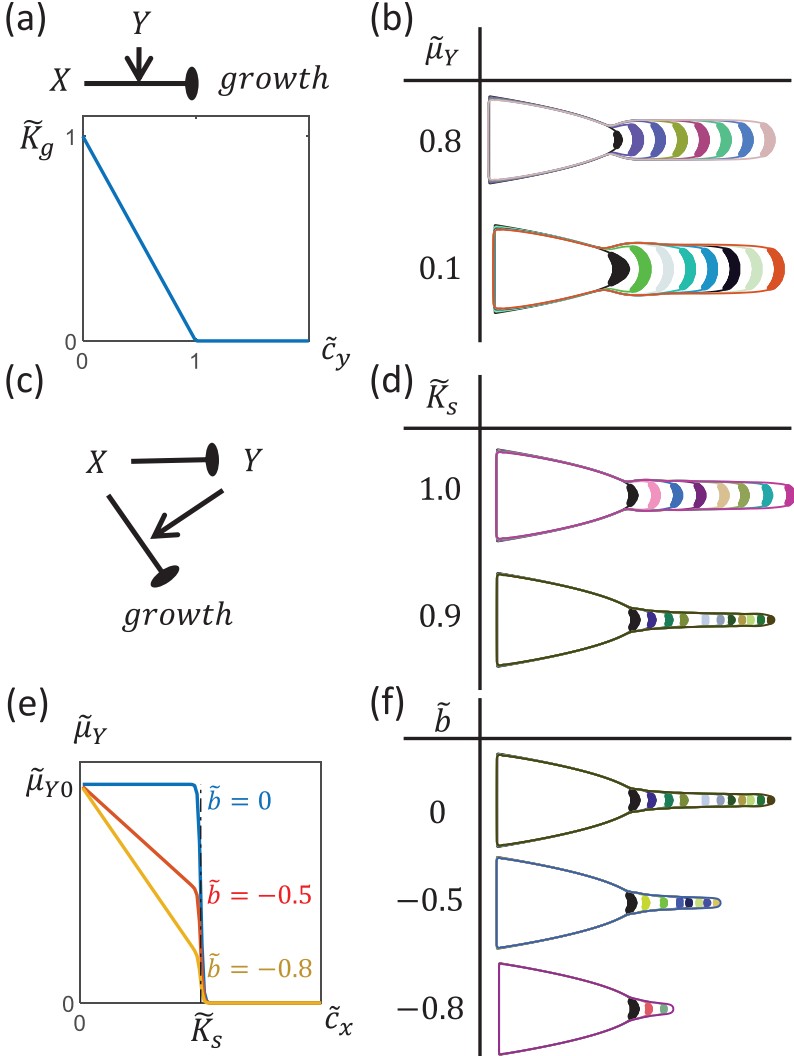

**Fig 5. Dynamics of cluster when cells secrete a growth threshold regulator in addition to a growth inhibitor.** (a) $Y$ reduces the growth threshold of $X$. Here, we have chosen the rescaled growth threshold $\tilde{K}_g = \frac{K_g}{K_{g0}}$ to decrease linearly with the rescaled concentration $\tilde{c}_y = ac_Y$. (Eq 9) (b) The cluster grows a rod-like structure when all cells secrete $Y$ at a constant effective rate $\tilde{\mu}_Y = \frac{a\mu_Y}{\gamma_1}\frac{D_1}{D_2}$. (c) We consider the scenario where $X$ inhibits the production of $Y$ such that only a region at the tip of the tissue can secrete $Y$. (d) For the scenario described in (c), it is possible for the protrusion to become narrower over time if the secretion threshold is less than the growth threshold $\tilde{K}_s = \frac{K_s}{K_{g0}} < 1$ (top: $\tilde{\mu}_Y = 6$, bottom: $\tilde{\mu}_Y = 15$). (e) We also allow the maximum secretion rate of $Y$ to decrease linearly with $c_X$. (Eq 10) (f) When $\tilde{K}_s = 0.9$, a stronger regulation of $\mu_{Y,max}$ (more negative $\tilde{b}$) produces sharper cone-like structures (top: $\tilde{\mu}_Y = 15$, middle: $\tilde{\mu}_Y = 26.5$, bottom: $\tilde{\mu}_Y = 48$). [Other dimensionless parameters: $\tilde{\mu}_X = 8$, $\gamma_r = 0.2$.].

S6 Fig). The protrusion eventually stops growing when the the area of the growth zone vanishes.

The length of the cone-like protrusion can be controlled by regulating the rate at which the size of the growth zone decreases. We find that this can be achieved through regulating the maximum secretion rate $\mu_{Y,max}$ of $Y$ in the regime where $Y$ is being secreted (i.e. $\tilde{c}_X < \tilde{K}_s$).

More specifically, we allow $\mu_{Y,max}$ to increase with decreasing $c_X$ (Fig 5E):

$$\tilde{\mu}_{Y,max} = \tilde{\mu}_{Y0}\max((1 + \tilde{b}\tilde{c}_X), 0), \tag{10}$$

where as before the overhead $\sim$ indicates the corresponding rescaled variables, and $\tilde{b} \leq 0$ is a dimensionless parameter that controls how strongly $X$ regulates $\mu_{Y,max}$. We find that a stronger regulation of $\mu_{Y,max}$ can give sharper cones (Fig 5F). This is because as the protrusion grows, $c_X$ decreases. A more negative $\tilde{b}$ therefore results in a greater increase in the production of $Y$, and hence a more drastic shrinkage of the growth zone as the tissue elongates (S6 Fig).

**An independent threshold-regulator should be used to control the growth of each set of protrusions.** If there are multiple protrusions such as in the presence of multiple growth inhibitors, we would expect that having a single growth threshold regulator would affect all protrusions in a similar or correlated way. Therefore, to control the lengths of these protrusions independently, we should have a different growth-threshold regulator for each set of protrusions. This set of growth-threshold regulators can regulate the same growth inhibitor, but their secretion rates (including the condition under which they are secreted) and their effect on the growth threshold may vary. Therefore, each regulator may be active only in the chemical environment of the protrusion it controls. Note however that due to the parabolic symmetry in our problem, the side protrusions can only be controlled in pairs.

## Discussion

There are many possible ways by which tissues can change their shape during the developmental process. In the context of tissue elongation [39], this can occur through localized proliferation such as in bird beaks [25], oriented cell divisions such as in the *Drosophila* wing disc epithelium [15], cell intercalation [12, 13, 40], and elongation of individual cells [39]. Many of these mechanisms fundamentally require cells to sense their local environment and respond by varying gene expression levels, which in turn regulate the growth rate of cells, the production rate of various intracellular proteins, the secretion rate of diffusible signals, the polarization of cells, amongst others. How these responses can be programmed, and how complicated the regulatory mechanisms need to be, to achieve a desired tissue shape and structure is a fundamental question in biology.

Here we explore changes in shape that arise solely from differential growth rates across the tissue, and investigate this in the context of growth rate regulation via diffusible morphogens. To do so, we adopt a framework which couples possible growth regulatory mechanisms to the overall expansion of the tissue. Given any regulatory circuit, the spatial chemical concentration profiles are modelled using the standard reaction-diffusion equations, and the corresponding growth rate of individual cells are specified as functions of these concentrations. The growth of the overall cell cluster is then obtained by treating it as an incompressible fluid, an approach that has been previously proposed and used to model a wide variety of systems ranging from the development of multicellular tissues [28–34] to the expansion of bacterial colonies [35–37]. The purpose of many of these studies is to recapitulate and hence understand a specific observation in a system. This requires incorporating details specific to that system-of-interest. For example, models of chick limb development involve specifying distinct regions (i.e., containing different cell types) within the tissue, namely the zone of polarizing activity and the apical epidermal ridge, that are separately responsible for producing the diffusible proteins Sonic Hedgehog (Shh) and fibroblast growth factors (FGFs) respectively [28, 29]. The concentrations of Shh and FGFs affect both the secretion rates of each other, and the proliferation rate of cells [28, 29]. In contrast, our goal here is to use this simple class of models to explore what is possible and how complex the regulatory schemes need to be when there is just a single cell type

whose growth rate is regulated by diffusible morphogens alone. With this bottom-up approach, we find with an initial parabolic cell cluster that it is possible to grow a rod-like extension with just a single growth inhibitor, and that having multiple growth inhibitors allows for multiple protrusions while growth threshold regulators can be used to regulate the shape and length of each protrusion, allowing protrusions to be cone-shaped rather than rod-like. In addition to what is achievable with chemicals with different functions, the limits of each regulatory circuit can also be inferred from these results given a fixed initial condition. In our example, a single growth inhibitor alone (with or without growth threshold regulators) cannot give rise to multiple protrusions. More generally, with the regulatory schemes studied here, the maximum possible number of protrusions grows exponentially with the number of growth inhibitors involved, and there should be an additional growth-threshold regulator for each set of protrusions one wishes to control independently. These results provide a lower bound for the number of signals cells need to achieve a certain goal. Presumably similar analyses and general arguments can be made for other structural features and the corresponding regulatory mechanisms. Furthermore, there may be multiple sets of rules that could give rise to qualitatively similar structural features. Such an analysis can therefore also be useful for comparing different regulatory schemes in terms of their capacity to generate complex structures.

Turing-like instabilities have been used to explain pattern formation in many biological systems and can give rise to digit patterning [41]. Even though Turing patterns may be a convenient way of getting a large number of protrusions with very few chemicals, they typically operate over narrow parameter ranges, and the nature of the Turing mechanism suggests that the type and features of the patterns may change drastically as the tissue grows and changes its shape. Coupling such a mechanism to growth regulation therefore entails another degree of complexity that would probably require fine-tuning of the parameters. Here, we take a different approach and our results show that it is possible to obtain multiple protrusions with other regulatory schemes that do not involve such spontaneous instabilities.

Identifying possible regulatory mechanisms and the minimum number of signals needed for achieving a certain structural feature may provide insight into natural developmental systems, and is especially useful for engineering these clusters synthetically. Given the advancement in genetic engineering techniques, the concrete predictions we have could be tested in the lab in the future. More importantly, our framework could potentially also be used to guide the design of regulatory circuits for achieving a desired target structure. In particular, this model could be used to test if a proposed/hypothesized regulatory scheme can achieve a particular structure and if so, what are the relevant parameter regimes. Furthermore, even though we have assumed that all cells in the cluster follow the same set of rules, our framework may potentially be extended to include multiple cell types, with each cell type following a different growth rule, or other processes such as cell differentiation (where cells change from one type to another based on local rules) and cell reorganization driven by differential adhesion.

Our framework can also set the basis for studying the process of regeneration, or self-repair. Considering the examples we have discussed in this paper, the regeneration dynamics of a cleaved protrusion depends on the cleaved surface. For example, when cells secrete a single growth inhibitor, a straight cut perpendicular to the elongation axis of the protrusion gives rise to two protrusions due to non-growth-inhibited cells at the corners of the cut (panel a in S7 Fig), while a curved cut that protrudes in the middle allows regeneration of a single protrusion (panel b in S7 Fig). This is consistent with the observation that during the growth of the single protrusion, the tip of the protrusion remains curved throughout growth. What types of regulatory circuits promote regeneration and what conditions are required for regeneration is an interesting question we leave for future work.

## Materials and methods

All simulations are carried out using custom code written in MATLAB (R2019a). These codes can be found in the GitHub repository: https://github.com/yipeiguo/ProgrammingGrowth.

## Supporting information

**S1 Table. List of model parameters, their definitions and their values used in the simulations.**
(PDF)

**S1 Text. Solving for velocities at tissue boundary.**
(PDF)

**S2 Text. Example of a regulatory mechanism involving a growth activator.**
(PDF)

**S1 Fig. Growth dynamics with an initial elliptic cluster for the various regulatory schemes.**
(a) Single growth inhibitor [parameters: $\tilde{\mu}_X = 7$], (b) 2 growth inhibitors [parameters: $\tilde{\mu}_X = 7$, $\tilde{\mu}_{Y0} = 59$, $\tilde{K}_s = 1$, $\gamma_r = 1$], and (c) 1 growth inhibitor and 1 growth-threshold regulator [parameters: $\tilde{\mu}_X = 7$, $\tilde{\mu}_Y = 50$, $\tilde{K}_s = 0.9$, $\gamma_r = 0.2$, $\tilde{b} = -0.8$]. [Other parameters: initial tissue length $\tilde{x}_0 = 1.5$, initial tissue width $\tilde{y}_0 = 1$].
(PDF)

**S2 Fig. A regulatory scheme involving a growth activator.** (a) All cells secrete $Y$, which inhibits the secretion of a growth activator $X$. (b) Example of growth dynamics with this regulatory scheme giving rise to a protrusion. [parameters: $\tilde{\mu}_{X0} = 48$, $\tilde{\mu}_Y = 7/1.2$, $\gamma_r = 0.5$, initial tissue length $\tilde{x}_0 = 1.5$, initial tissue width $\tilde{y}_0 = 1$ (see S2 Text for details of model description)].
(PDF)

**S3 Fig. Illustration of a protrusion formation process for cells producing a single growth inhibitor.** Starting with a small elliptical cluster, initially none of the cells are inhibited (red regions represent growth zones) and the cluster expands in all directions (boundary velocities indicated by the thickness of the blue region perpendicular to the tissue surface). As the number of cells increases, some of the cells stop dividing as concentration of the growth inhibitor exceeds a threshold. The tissue eventually elongates only along the horizontal axis as the growth zone becomes restricted to the tips of the cluster. [Parameters: $\mu_{eff} = 7$, initial tissue length $\tilde{x}_0 = 1.6$, initial tissue width $\tilde{y}_0 = 0.4$.].
(PDF)

**S4 Fig. Illustration of concentration profile of growth inhibitor $Y$.** With the 2 growth inhibitors regulatory scheme (in Fig 4a), for a fixed $\tilde{\mu}_X = 8$, the shape of the growth zone depends on the concentration profile of growth inhibitor $Y$. The background colors (with the corresponding legends) represent $\tilde{c}_Y$. Growth is inhibited when $\tilde{c}_Y > 1$, as can be seen from the shapes of the growth zones (indicated by the regions in red). [Other parameters: $\gamma_r = 1$, initial tissue length $\tilde{x}_0 = 1.5$, initial tissue width $\tilde{y}_0 = 1$.].
(PDF)

**S5 Fig. Illustration of how the tissue grows multiple protrusion when there are two growth inhibitors.** [Parameters: $\tilde{\mu}_X = 8$, $\tilde{\mu}_Y = 65$, $\tilde{K}_s = 1$ $\gamma_r = 1$, initial tissue length $\tilde{x}_0 = 1.5$, initial tissue width $\tilde{y}_0 = 1$.].
(PDF)

**S6 Fig. How features of the tissue change as a function of its elongation $\Delta x$ during growth of a cone-like structure with the regulatory scheme of having 1 growth inhibitor $X$ and 1 growth-threshold regulator $Y$ (Fig 5c) for different values of $\tilde{b} = 0$ (red circles), $\tilde{b} = -0.5$ (green squares), $\tilde{b} = -0.8$ (purple triangles) (see Eq 10).** (a) As the tissue elongates, the minimum concentration $c_{X,min}$ of $X$ at the tissue tip decreases. (b) Since $X$ inhibits the secretion of $Y$, a reduction in $c_X$ increases the area $A_Y$ within which cells are secreting $Y$. (c) As $A_Y$ increases, $c_Y$ at the tissue tip also increases. Since a higher $|\tilde{b}|$ implies a larger increase in the secretion rate $\tilde{\mu}_{Y,max}$ of $Y$ as $c_X$ decreases, $c_Y$ increases faster when $|\tilde{b}|$ is larger. (d) Since $Y$ reduces the growth threshold of $X$, an increase in $c_Y$ reduces the area $A_{gz}$ of the growth zone, with a faster decrease when $|\tilde{b}|$ is larger. [Other parameters: $\tilde{\mu}_X = 8$,

$$\tilde{\mu}_Y = \begin{cases} 15, & \text{when } \tilde{b} = 0 \\ 26.5, & \text{when } \tilde{b} = -0.5 \\ 48, & \text{when } \tilde{b} = -0.8 \end{cases}, \tilde{K}_s = 0.9 \; \gamma_r = 0.2, \text{ initial tissue length } \tilde{x}_0 = 1.5, \text{ initial tissue}$$

width $\tilde{y}_0 = 1$.].
(PDF)

**S7 Fig. Illustration of a regeneration process of a cluster of cells secreting a single growth inhibitor.** (a) With a straight cut (red vertical line) perpendicular to the protrusion (left), the cluster grows two protrusions (right). This arises because compared to the original rounded edge of the protrusion, there are more cells at the corner and they are uninhibited by the growth inhibitor (middle). Red region indicates the growth zone. (b) WIth a curved (elliptical) cut (left), a single protusion reemerges (right) from a growth zone at the tip of the protrusion (middle). [Parameters: $\tilde{\mu}_X = 8$.].
(PDF)

## Author Contributions

**Conceptualization:** Yipei Guo, Mor Nitzan, Michael P. Brenner.

**Formal analysis:** Yipei Guo.

**Investigation:** Yipei Guo.

**Supervision:** Michael P. Brenner.

**Visualization:** Yipei Guo.

**Writing – original draft:** Yipei Guo.

**Writing – review & editing:** Yipei Guo, Mor Nitzan, Michael P. Brenner.

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
