## [Decision Letter · Decision Letter 0]

26 May 2021

Dear Ms Guo,

Thank you very much for submitting your manuscript "Programming cell growth into different cluster shapes using diffusible signals" for consideration at PLOS Computational Biology.

As with all papers reviewed by the journal, your manuscript was reviewed by members of the editorial board and by several independent reviewers. In light of the reviews (below this email), we would like to invite the resubmission of a significantly-revised version that takes into account the reviewers' comments.

We cannot make any decision about publication until we have seen the revised manuscript and your response to the reviewers' comments. Your revised manuscript is also likely to be sent to reviewers for further evaluation.

Sincerely,

Alexandre V. Morozov, Ph.D.

Associate Editor

PLOS Computational Biology

Natalia Komarova

Deputy Editor

PLOS Computational Biology

Reviewer's Responses to Questions

**Comments to the Authors:**

Reviewer #1: The manuscript by Guo et al. presents a modeling framework for the analysis of tissue growth patterning in a concentration vector field of several interacting chemical species regulating tissue growth. Essentially, the approach treats the tissue as a continuous field, with local differences in growth rate that is determined by superposition of all growth factors, which are described by a standard reaction-diffusion model. The authors then restrict themselves to the analysis of models with one or several growth inhibitors, and find that growth inhibitors can induce growth of protrusions, the number of protrusions scaling exponentially with the number of growth inhibitors.

The authors present an innovative modeling framework, and therefore the paper seems appropriate for the journal. However, the analysis and results sections fall somewhat short, especially in light of the broad questions and claims suggested in the abstract and introduction. For proof of principle of the modeling framework, a minimal requirement would be an example with a growth stimulator, and a straight-forward case would also be a combination of a growth inhibitor and a growth stimulator with varying degree of interaction. Furthermore, it would be appropriate to spell out at least one of the examples mentioned in the introduction (notch receptor, Hedgehog secretion, synthetic activator-inhibitor system) in terms of a more specific model.

Further comments:

p.2, “we model the cell cluster as an incompressible cellular fuid of constant density…” – really? From my understanding, Eq 4 is the opposite of an incompressible fluid which is defined by vanishing divergence. Would Eq. 4 not rather suggest an analogy to an electric field where g would correspond to the charge density?

p. 3, “Inspired by the example of developing bird beaks … we choose to initialize the system with a 2-dimensional parabolic cell cluster” – not clear why bird beaks should be the reference point here. Could the authors discuss the choice of the region and initial conditions in light of the examples given in the introduction? Furthermore, a model scheme illustrating the simulated region and starting conditions would be very helpful.

Reviewer #2: Guo et al, have made use of simulation studies in order to provide insights into the complexity of signals required and how they can be programmed to enable shape formation by tissues. They look into the case of morphogen gradients and growth-based shape formation. Their simulation studies indicate that it is possible to form a rod-shaped structure using just one growth inhibitory signal. In addition, they show that using more than one growth inhibitory signals, it is possible to allow the formation of several protrusions. Finally, they illustrated that the shape and length of each protrusion can be influenced by threshold regulators. This is an important topic and the paper and approach taken are of interest for the emerging field of multicellular systems engineering.

Please find my comments below:

1) While the model has attempted to explain process of elongation (initiation of protrusion, numbers and also forms), there is less information about possible ways to initiate and terminate the formation of protrusion. This can be discussed, and possible models can be provided in 3 sets:

a) How to terminate formation of morphogenesis ?

b) How to initiate the process? how the inhibitory access is established in short range

c) How the process is maintained (and moves forward)? (perhaps this is the part that is already covered)

2) Would the model support “regeneration” of protrusion after injury? In discussion please also add how the model may explain how/why regeneration can fail in some cases of tissue injury or in some species.

3) Line 1 to 3: Synthetic biology is a broad term, and it would be more accurate to say: A fundamental goal of synthetic developmental biology is to understand how cells can be programmed to generate a desired spatial configuration.

4) For easier read, the authors are expected to clearly annotate all the parameters used in their equations in a consistent manner and provide a clear word description of each of the parameters used (perhaps in a separate table).

5) Authors mention :” we illustrate how our approach can be used to guide the design of regulatory circuits for achieving a desired target structure”. Please elaborate in abstract and in text about this notion. It is not clear if they really achieve this. Perhaps either tune down the claim or provide more explanation for readers to truly prove ability to guide the design. Unfortunately since there is no wet lab experiment to show accuracy of prediction this calim should be done carefully.

6) The authors are expected to compare and contrast their proposed models with other previously established models in the discussion section. What are the strength and weaknesses that each model has and what part of in vivo processes are not met with current models. Additionally, for example: they build up their model based on the tip proliferating cells (distal cells) that grow and become farther from the source inhibitory signal. Would it be possible to think about proximal cells (at base) to grow and promote elongation and an activator signal that is produced by more distal cells.

**Have the authors made all data and (if applicable) computational code underlying the findings in their manuscript fully available?**

Reviewer #1: Yes

Reviewer #2: Yes

PLOS authors have the option to publish the peer review history of their article (what does this mean?). If published, this will include your full peer review and any attached files.

Reviewer #1: No

Reviewer #2: No
---

## [Decision Letter · Decision Letter 1]

24 Sep 2021

Dear Ms Guo,

Thank you very much for submitting your manuscript "Programming cell growth into different cluster shapes using diffusible signals" for consideration at PLOS Computational Biology. As with all papers reviewed by the journal, your manuscript was reviewed by members of the editorial board and by several independent reviewers. The reviewers appreciated the attention to an important topic. Based on the reviews, we are likely to accept this manuscript for publication, providing that you modify the manuscript according to the review recommendations.

Sincerely,

Alexandre V. Morozov, Ph.D.

Associate Editor

PLOS Computational Biology

Natalia Komarova

Deputy Editor

PLOS Computational Biology

[LINK]

Reviewer's Responses to Questions

**Comments to the Authors:**

Reviewer #1: The paper improved substantially in revision, and the authors have addressed most of my points. Especially, adding the case without growth regulation and adding more references to previous work was very helpful. However, I am still somewhat confused about the scope of the paper, and I would ask the authors to clarify that before publication.

In particular, the authors wrote in response to my first review:

“We now describe in the introduction developing bird beaks as an example of how diffusible morphogens regulate a shape of a tissue. This contrasts with the other examples of Hedgehog-secreting cells, (…)”.

On the other hand, in the revised introduction, there is no clear distinction between the two types of examples, nor are any conceptual differences pointed out. Rather, the introduction still gives the impression that all mentioned examples can be treated with the theoretical concept presented in the manuscript. I now understand that this is not the case, which should be clearly stated and discussed.

Furthermore, I would expect already in the introduction a reference to previous work using similar concepts (such as lines 605-609 in the discussion), as well as a statement regarding the “goal” (line 622), the novelty and the scope of the presented approach.

Reviewer #2: -

**Have the authors made all data and (if applicable) computational code underlying the findings in their manuscript fully available?**

Reviewer #1: Yes

Reviewer #2: **No: **it is not fully clear . however the authors point out all data are available in supplementary file.

PLOS authors have the option to publish the peer review history of their article (what does this mean?). If published, this will include your full peer review and any attached files.

Reviewer #1: No

Reviewer #2: No

Figure Files:

Data Requirements:

Reproducibility:

References:

---

## [Editor Report · Decision Letter 2]

19 Oct 2021

Dear Dr Guo,

We are pleased to inform you that your manuscript 'Programming cell growth into different cluster shapes using diffusible signals' has been provisionally accepted for publication in PLOS Computational Biology.

Best regards,

Alexandre V. Morozov, Ph.D.

Associate Editor

PLOS Computational Biology

Natalia Komarova

Deputy Editor

PLOS Computational Biology

---

## [Editor Report · Acceptance letter]

3 Nov 2021

PCOMPBIOL-D-21-00128R2 

Programming cell growth into different cluster shapes using diffusible signals

Dear Dr Guo,

I am pleased to inform you that your manuscript has been formally accepted for publication in PLOS Computational Biology. Your manuscript is now with our production department and you will be notified of the publication date in due course.

With kind regards,

Zsofia Freund
